# Transient Receptor Potential Melastatin 2 (TRPM2) Inhibition by Antioxidant, *N*-Acetyl-l-Cysteine, Reduces Global Cerebral Ischemia-Induced Neuronal Death

**DOI:** 10.3390/ijms21176026

**Published:** 2020-08-21

**Authors:** Dae Ki Hong, A Ra Kho, Song Hee Lee, Jeong Hyun Jeong, Beom Seok Kang, Dong Hyeon Kang, Min Kyu Park, Kyoung-Ha Park, Man-Sup Lim, Bo Young Choi, Sang Won Suh

**Affiliations:** 1Department of Physiology, College of Medicine, Hallym University, Chuncheon 24252, Korea; zxnm01220@gmail.com (D.K.H.); rnlduadkfk136@hallym.ac.kr (A.R.K.); sshlee@hallym.ac.kr (S.H.L.); jd1422@hanmail.net (J.H.J.); ttiger1993@gmail.com (B.S.K.); bagmingyu50@gmail.com (M.K.P.); 2Department of Medical Science, College of Medicine, Hallym University, Chuncheon 24252, Korea; ehdgus6312@gmail.com; 3Division of Cardiovascular Diseases, Hallym University Medical Center, Anyang 14068, Korea; pkhmd@naver.com; 4Department of Medical Education, College of Medicine, Hallym University, Chuncheon 24252, Korea; ellemes@hallym.ac.kr

**Keywords:** global cerebral ischemia, *N*-acetyl-l-cysteine, transient receptor potential melastatin 2, zinc, neurodegeneration

## Abstract

A variety of pathogenic mechanisms, such as cytoplasmic calcium/zinc influx, reactive oxygen species production, and ionic imbalance, have been suggested to play a role in cerebral ischemia induced neurodegeneration. During the ischemic state that occurs after stroke or heart attack, it is observed that vesicular zinc can be released into the synaptic cleft, and then translocated into the cytoplasm via various cation channels. Transient receptor potential melastatin 2 (TRPM2) is highly distributed in the central nervous system and has high sensitivity to oxidative damage. Several previous studies have shown that TRPM2 channel activation contributes to neuroinflammation and neurodegeneration cascades. Therefore, we examined whether anti-oxidant treatment, such as with *N*-acetyl-l-cysteine (NAC), provides neuroprotection via regulation of TRPM2, following global cerebral ischemia (GCI). Experimental animals were then immediately injected with NAC (150 mg/kg/day) for 3 and 7 days, before sacrifice. We demonstrated that NAC administration reduced activation of GCI-induced neuronal death cascades, such as lipid peroxidation, microglia and astroglia activation, free zinc accumulation, and TRPM2 over-activation. Therefore, modulation of the TRPM2 channel can be a potential therapeutic target to prevent ischemia-induced neuronal death.

## 1. Introduction

Recent decades have seen a dramatic increase in the attention and focus on the study of brain disease. In particular, cerebral ischemia is a global problem and is the leading cause of brain disease and morbidity [1,2]. There are many reasons why cerebral ischemia is so widespread, in part due to the diverse number of individual pathologies that converge on the emergence of the ischemic state; the largest risk factor for development of cerebral ischemia is occlusion of blood vessels to the brain. Upon the occlusion of these blood vessels, cerebral blood flow is considerably reduced to regions of the brain and essential supplies of oxygen, nutrients, glucose, and many other cellular substrates are reduced [3]. Cerebral ischemia is divided into two sub-categories. The first, focal cerebral ischemia, involves transiently or permanently reduced blood flow and nutrient supply to particular brain areas that are directly fed by the occluded vasculature. Damage to the brain after focal ischemia is thus focused on the affected areas and injury occurs locally [4]. Global cerebral ischemia (GCI) is similar to focal cerebral ischemia, although the mechanisms of induction and patterns of damage to specific brain regions are different. This brain injury occurs due to several reasons, but is best described by a sudden and near-complete stop in cerebral blood-flow, such as is seen under cardiac arrest-induced cessation of blood circulation [5].

Once the ischemic insult has occurred, abnormal physiological changes to brain function have already began to occur. Under the ischemic condition, disruption of brain homeostasis is disturbed and cell death mechanisms are initiated. This cerebral injury leads to an alteration in ionic balance within the intra-extracellular space [6], reactive oxygen species (ROS) production [7], microglia [8] and astrocyte [9] activation, zinc accumulation [10], and neuronal cell degeneration [11,12,13] in the brain. Blockage of blood flow to the brain by ischemic insult causes tissue infarction, dysfunction of ionic balance, and intracellular calcium overload [14]. In a previous study, excessive release of calcium ions from synaptic vesicles after brain ischemia was shown to have deleterious effects on neurons and glia [15]. Overloaded intracellular calcium ions trigger glutamate release, proteolysis, mitochondrial dysfunction, and other deleterious cascades [16]. As discussed above, calcium overload contributes to ischemic injury-induced cell death mechanisms. According to a recent report, the cation zinc has an essential role in brain ischemia, in addition to the calcium ion [17].

Zinc is the most widely distributed metal ion in the central nervous system, and is especially abundant in the hippocampus. Zinc has many important roles in enzymes and transcription factors, and its concentration in neurons and glia is controlled by binding proteins, known as metallothionein, as well as several transporters and channels [17,18,19]. In addition, zinc regulates neural synaptic transmission, innate and adaptive immunity, signal transduction, and cell proliferation, particularly for DNA synthesis [20,21]. In the pre-synaptic vesicles, chelatable zinc is abundant. It is released into the synaptic cleft, moved into pre-synaptic neurons via several transporters, and subsequently undergoes reuptake from the synaptic cleft under normal physiological conditions [22,23]. Under abnormal physiological conditions, such as neurological disorders, zinc homeostasis is disturbed and dysregulated. In the case of neurological disorders, vesicular zinc is excessively released and largely accumulates within post-synaptic neurons. In addition to releasing zinc from synaptic vesicles, zinc that is loosely bound by proteins can be divided into the categories of chelatable free zinc and proteins [12,24,25]. Pathological zinc accumulation can influence cellular dysfunction in several ways. In a previous study, it was demonstrated that zinc-treated cultured neurons produce ROS from their mitochondria [18] and trigger microglial activation [26]. In addition, previous studies by our group have observed that excessive zinc accumulation within brain hippocampal regions can lead to neurodegeneration after cerebral insult, such as ischemia [10,27], head trauma [25], hypoglycemia [28], and cultured neurons [29].

*N*-acetyl-l-cysteine (NAC), which contains an acetylated cysteine residue, acts as a powerful antioxidant by supplementation of cysteine and promotion of glutathione biosynthesis [30,31]. Injury-induced oxidative damage plays a primary role in central nervous system dysfunction and contributes to neurodegenerative cascades [32,33]. To protect against brain injury-induced oxidative damage, our group has previously used NAC to treat hypoglycemia and traumatic brain injury. NAC stabilized glutathione concentrations, reduced degenerating neurons, and decreased oxidative damage within hippocampal regions [24,34]. Hippocampal neurodegeneration can be prevented or reduced when oxidative damage and other deleterious cascades are prevented by regulating physiologic concentrations of glutathione via providing ample cysteine, which is supplied from NAC [35]. Moreover, NAC has binding sites for trace metals that include copper, magnesium, and zinc. These studies demonstrate that NAC can neutralize toxically accumulated trace metals occurring in neuropathies [24,36].

Transient receptor potential melastatin 2 (TRPM2) is one of the non-selective cationic channels, and the activity of this channel is triggered by ROS production promoted by deleterious cell death mechanisms in several neurological disorders. Recently, the TRPM2 channel has been reported to be included in both physiological and pathological conditions [37,38]. Several previous studies have shown that the TRPM2 channel is expressed in the central nervous system (CNS), which is largely expressed within the hippocampus, stratum, and cortex [39]. These studies found that several divalent cations, for example, calcium and magnesium, moved into the intracellular space via TRPM2, and that its activity was regulated by external stimuli such as ROS, hydrogen peroxide, and tumor necrosis factor-α (TNF-α) [40,41,42]. In addition, TRPM2 is distributed in both neurons and glia, and is involved in pathophysiological conditions of nitric oxide (NO) and ROS production signaling [38,43]. Following several neurological injuries, excessive calcium influx via TRPM2 can trigger release of diverse cytokines and thus contributes to the inflammatory response [44]. Several brain diseases, such as stroke or head trauma, lead to excitotoxicity, which is caused by the inappropriate activation of voltage-dependent calcium channels and excitatory amino acid release from the dendritic and presynaptic space into the extracellular cleft. Previous studies verified that in a rodent disease model of ischemia, TRPM2 mRNA levels were significantly increased. Interestingly, a recent study demonstrated that TRPM2 deficiency in mice decreased cytosolic zinc concentrations and reduced zinc influx from the extracellular space [45]. These findings suggest that, during several brain diseases, TRPM2 is activated and this contributes to cell death mechanisms [44,46].

In the present study, we tested the hypothesis that if ischemia-induced oxidative damage leads to TRPM2 activation, inhibition of this channel may have therapeutic effects through blockade of zinc influx-induced cell damage, which is supported by evidence that NAC administration decreased brain damage in several other neurological diseases [24,34]. Following this logic, we hypothesized that if the TRPM2 level was increased by global cerebral ischemia-induced cell death cascades, regulation of TRPM2 as a potential target for preventing ischemia-induced injury could ameliorate a diverse number of cerebral pathologies associated with ischemia. In addition, we speculated that the zinc ion is closely related to TRPM2 and is excessively moved into the intracellular space via this channel. To test our hypothesis, we used a GCI brain disease model in adult rats and conducted several types of histological evaluations and behavioral assessments.

## 2. Results

### 2.1. Global Cerebral Ischemia-Induced Neuronal Death and Zinc Accumulation Is Attenuated by Post-Administration of NAC

Timeline showing the experimental design for 3 days and 7 days (Figure 1A). Whole GCI conducted experimental animals were monitored for arterial blood pressure and electroencephalography continuously during the before-isoelectric, isoelectric, and after-blood reperfusion condition (Figure 1B–D).

To determine whether GCI-induced neuronal death and zinc accumulation was attenuated by NAC post-administration, we used FJB (Fluoro-Jade B) and TSQ (*N*-(6-methoxy-8-quinolyl)-para-toluenesulfonamide staining at 3 days after GCI. The degenerating neurons were strongly present in the GCI–vehicle groups. When compared with vehicle-treated groups, NAC-treated groups showed significantly reduced numbers of degenerating neurons (Figure 2A). Quantified numbers of degenerating neurons are displayed using bar graphs (vehicle: subiculum, 168.48 ± 19.63; CA1, 140.03 ± 16.81; CA2, 132.75 ± 14.96; NAC: subiculum, 52.17 ± 20.09; CA1, 54.97 ± 19.74; CA2, 54.98 ± 19.24; Figure 2B). Consequently, GCI-induced zinc accumulation in post-synaptic neurons was observed in vehicle-treated groups. However, NAC administration after GCI reduced zinc accumulation in hippocampal areas (Figure 2C). Representative bar graph shows number of quantified TSQ-positive cells in both GCI–vehicle and GCI–NAC groups (vehicle: subiculum, 84.6 ± 4.29; CA1, 76.28 ± 5.32; CA2, 76.53 ± 3.18; NAC: subiculum, 37.66 ± 2.18; CA1, 32.63 ± 10.52; CA2, 27.77 ± 4.77; Figure 2D).

### 2.2. NAC Post-Treatment Regulates Global Cerebral Ischemia-Induced Loss of Neuronal Glutathione and Lipid Peroxidation

One strategy to solve the problem of ischemic injury is to focus on reducing oxidative damage. We speculated that post-administration of NAC can supply cysteine to produce the powerful anti-oxidant glutathione and influence neuronal oxidative damage. Representative images show glutathione (GS-NEM) levels (Figure 3A). Sham groups show a wide distribution of glutathione within neurons, but the GCI–vehicle group shows a loss of glutathione after injury, particularly oxidative damage. NAC treatment after GCI elevates glutathione concentration in damaged neurons. Based on the above results, we confirmed whether NAC administration directly protects against ROS-induced neuronal oxidative damage by enhancing glutathione levels. ROS triggers lipid peroxidation, and 4-hydroxy-2-nonenal (4-HNE) is a histological marker for lipid peroxidation. GCI led to an increase in ROS, and 4-HNE-positive fluorescence signal was largely increased in the hippocampal subiculum, CA1, and CA2 regions (Figure 3B). Increased lipid peroxidation by GCI-induced ROS production was significantly attenuated following post-treatment of NAC (mean gray value, GCI–vehicle: subiculum, 25.93 ± 2.08; CA1, 28.89 ± 1.69; CA2, 26.36 ± 2.41; GCI–NAC: subiculum, 17.20 ± 0.97; CA1, 16.34 ± 0.99; CA2, 16.90 ± 0.31; Figure 3C). Sham groups had no differences in 4-HNE signal (sham–vehicle: subiculum, 11.12 ± 0.37; CA1, 9.76 ± 0.64; CA2, 11.12 ± 0.97; sham NAC: subiculum, 10.10 ± 0.92; CA1, 8.65 ± 0.43; CA2, 10.89 ± 1.21; Figure 3C).

### 2.3. NAC Attenuates TRPM2 Activation in Hippocampal Neurons and Glial Cells

GCI activates TRPM2 channels in the brain, which is localized to glial cells in addition to hippocampal neurons. To demonstrate whether TRPM2 is located within neurons or glial cells in our experimental setting, we conducted immuno-fluorescence staining. Figure 4A shows distribution of TRPM2 channels in the hippocampal pyramidal layer CA1 region, where it was co-localized with neuronal nuclei (NeuN). There was no difference in TRPM2-positive fluorescence signals in sham groups (mean gray value, sham–vehicle: 11.08 ± 0.58; sham NAC: 11.09 ± 0.62; Figure 4B). However, the ischemic condition contributes to activation of TRPM2 in the brain and this is highly localized to the hippocampal pyramidal CA1 layer, which is also known to be especially vulnerable to ischemic injury. The TRPM2-positive signal was dramatically increased in the GCI–vehicle group and significantly decreased in the NAC post-treatment group (GCI–vehicle: 29.55 ± 1.15; GCI–NAC: 18.25 ± 1.76, Figure 4B).

To explore the localization of TRPM2 in non-neuronal cells, we conducted glial cell immunostaining. Sham groups showed that Iba-1-positive microglia were distributed throughout the hippocampus in an inactivated state. However, GCI triggers microglial activation, which was significantly decreased by NAC post-treatment (Figure 5A). The bar graph shows quantified microglia intensity (mean gray value, sham–vehicle, 7.59 ± 0.70; sham NAC, 7.82 ± 0.54; GCI–vehicle, 48.35 ± 4.60; GCI–NAC, 23.59 ± 1.01; Figure 5B). Figure 5C shows co-localized TRPM2 with microglia in GCI–vehicle groups. Similarly, astrocytes were distributed throughout the hippocampus, and there were no differences in sham groups. GCI triggers astrocyte activation, which was significantly reduced by NAC post-treatment (Figure 5D). The bar graph shows quantified astrocyte activation (mean gray value, sham–vehicle, 12.52 ± 1.29; sham NAC, 11.88 ± 1.96; GCI–vehicle, 61.46 ± 5.21; GCI–NAC, 36.75 ± 4.58; Figure 5E). Figure 5F shows co-localized TRPM2 with astrocyte in GCI–vehicle groups.

### 2.4. NAC Restores Global Cerebral Ischemia-Induced Sensorimotor Deficit, Neurologic Decline, and Neurodegeneration

To investigate the effects of NAC post-administration on GCI-induced sensorimotor deficit and loss of neurological function, experimental animals were analyzed using the adhesive removal test and modified neurological severity score (mNSS) methods. First, to evaluate sensorimotor deficit after ischemic damage we used the adhesive removal test method. The GCI–vehicle-treated groups spent longer periods of time recognizing adhesive tapes on palms than the GCI–NAC-treated groups after ischemic insult. Sham–vehicle and NAC groups had no difference in removal time during the test days. These results showed that GCI-induced sensorimotor deficit was restored by NAC treatment (Figure 6A). Next, we also assessed whether NAC administration restored GCI-induced neurological decline. Because the sham groups had no impairment, the mNSS count was zero. Following GCI, neurological impairment lead to high scores in the vehicle group and NAC administration attenuated this neurologic decline (Figure 6B). In addition, we determined whether post-administration of NAC provides neuroprotective effects on GCI-induced neurodegeneration following 7 days of consecutive treatment (Figure 6C). Sham groups show numerous neuronal nuclei (NeuN)-positive cells in hippocampal subiculum, CA1, and CA2 regions. There were no significant differences in the number of NeuN-positive cells between the sham–vehicle and NAC group (sham–vehicle: subiculum, 170.5 ± 5.5; CA1, 180.5 ± 5.5; CA2, 210 ± 11, sham-NAC: subiculum, 177 ± 7; CA1, 179 ± 10; CA2, 205 ± 7; Figure 6D). In contrast, the number of NeuN-positive cells was dramatically decreased in the GCI–vehicle group. However, NAC administration significantly restored the number of NeuN-positive cells in the hippocampal regions (GCI–vehicle: subiculum, 79.95 ± 7.69; CA1, 62.74 ± 11.15; CA2, 66.82 ± 4.91, GCI–NAC: subiculum, 110.47 ± 6.42; CA1, 108.12 ± 9.87; CA2, 110.31 ± 7.81; Figure 6D).

## 3. Discussion

In the present study, we verified whether NAC administration attenuates GCI-induced neuronal death via inhibition of TRPM2 channels. Our findings indicate that NAC reduces accumulation of intracellular free zinc and TRPM2 over-activation by maintaining a sufficient supply of glutathione production. Cysteine is the rate-limiting substrate required for the biosynthesis of glutathione and the cysteine residue present in NAC can serve as an immediate source of cellular cysteine to maintain high levels of intracellular glutathione, which in turn protects against oxidative damage arising from free radicals. In addition, daily administration of NAC for 3 days after GCI also clearly attenuated several GCI-induced cell death cascades involving neurodegeneration, intracellular free zinc accumulation, oxidative damage-induced lipid peroxidation, loss of glutathione, TRPM2 over-activation, and glial activation. Furthermore, post-treatment of NAC for 7 days prevented GCI-induced delayed hippocampal neuron death, sensorimotor impairment, and neurological deficits. Taken together, these results suggest that reduction of intracellular free zinc and restoration of neuronal GSH concentration by NAC administration attenuates GCI-induced neurodegenerative cascades and promotes behavioral impairments. Therefore, NAC may be an outstanding therapeutic agent for preventing GCI-induced neuronal death.

In general, ionic gradients are actively maintained within finely tuned ranges via several selective and non-selective ion channels throughout the central nervous system, and this process is fundamental to preserving healthy brain function. However, during many neurological injuries that include ischemic stroke [12] and traumatic brain injury [25] damage leads to an ionic imbalance, which in turn allows for the initiation of multiple neurodegenerative cascades. Our previous studies have focused on these mechanisms that follow ionic imbalance in several neurological diseases, such as global ischemia [13,47], hypoglycemia [48], and traumatic brain injury [49]. In particular, we have focused on the abnormal phenomenon of excessive release of vesicular zinc ions following brain injuries and its subsequent cellular pathology. Chelatable zinc that is liberated in this manner moves into post-synaptic neurons through non-selective cation channels, TRPM2 [50], and once chelatable zinc has accumulated within post-synaptic neurons it acts as a neurotoxin [51]. TRPM2 is one of the non-selective cation channels that regulates physiological and pathological processes in the central nervous system via modulation of numerous signaling pathways [38]. In addition, brain disease-induced ROS production triggers TRPM2 activation, leading to calcium influx from the extracellular space, and finally triggers cell death cascades that include PARP-1/PARG and caspase-dependent cell-death pathways [50,52,53]. In this present study, we focused on attenuating the effects of excessively released free zinc ions from synaptic vesicles after GCI that translocate into post-synaptic neurons via the TRPM2 channel. Hence, we hypothesized and verified that by reducing ROS production through supplementation of cysteine by NAC administration to maximize glutathione production, one can reduce hippocampal neurodegeneration through inhibition of zinc accumulation and TRPM2 activation.

*N*-acetyl-l-cysteine (NAC) is a derivative of cysteine and acts as a scavenger for elimination of free radicals [54]. In the present study, we focused on NAC as a cysteine donor to improve glutathione synthesis, and also as a chelator of heavy metal ions. Excessively released and accumulated free zinc ions after ischemic stroke have a deleterious effect on the nervous system. Previous studies determined that NAC can bind heavy metals (nickel, copper, zinc) because NAC has a thiol group (-SH), which allows it to bind cationic metal ions [55]. In addition, NAC can restore neuronal glutathione concentrations and protect against brain injuries. Kho et al. recently demonstrated that hypoglycemic brain disease-induced hippocampal neurodegeneration, lipid peroxidation, and zinc accumulation were restored to pre-insult levels by NAC administration [24]. Furthermore, Choi et al. recently verified that genetic deletion of the cysteine-related transporter, excitatory amino acid carrier 1 (EAAC1), exacerbates traumatic brain injury (TBI)-induced neurological damage, compared with wild type controls. These cell death cascades were attenuated by NAC treatment in both wild type and EAAC1 gene deletion animals [34]. In light of the abovementioned results, we hypothesized that NAC treatment after GCI may chelate zinc ions, and reduce accumulation of zinc-induced neurotoxicity and downregulation of the TRPM2 channel by reducing ROS production.

The post-global ischemic neuro-inflammatory response contributes to formation of the glial scar and exacerbates neuronal damage. Both microglia and astrocytes play an essential role in maintaining ionic gradients within the extracellular space and in supporting neurotransmission in neighboring neurons. In general, inactivated (resting) microglia and astrocytes regulate the maintenance of central nervous system homeostasis [56]. However, neurodegenerative diseases, such as head trauma, multiple sclerosis, epilepsy and stroke, trigger gliosis which is a non-specific alteration of glial cells and this can lead to glial scar formation if unchecked [57]. Lakhan et al. demonstrated that stroke-induced excitotoxicity and oxidative damage triggers activation of microglia and astrocytes, and these cells secrete several cytokines, chemokines, and matrix metalloproteases [58]. In addition, Kauppinen et al. verified that the zinc-overloaded condition alters the morphology of resting microglia to an activated form in cultured microglia, which was attenuated by treatment with the extracellular zinc chelator, CaEDTA, during the ischemic condition [59]. Barreto et al. reported that astrocyte viability was important for the restoration of neurons following ischemic damage. Furthermore, they asserted that although reactive astrocytes trigger the production of pro-inflammatory cytokines, they also perform functions for maintaining neuronal survival [9].

In the present study, our findings indicate that TRPM2 activation in neurons and glial cells were found after GCI, but were significantly correlated with astrocytes. Astrocytes have a dual nature when activated that can both alleviate or exacerbate the effects of brain damage depending on a sensitive mix of cellular factors. Accordingly, Barreto et al. assert that any solution that adequately addresses the problem of stroke-induced neurodegenerative must involve astrocytes [9]. No details about the interaction of TRPM2 and astrocytes has been reported, but we assumed that there must be some ischemia-related cascades between TRPM2 channels and astrocytes, and began to test this hypothesis. In the case of our research, TRPM2 activation in microglia was only mildly observed compared with astrocytes. However, microglia have a strong influence on CNS diseases, and it is also necessary to consider them for understanding injury-induced neuroinflammation and neurodegeneration in several neurological conditions. Malko et al. reported that calcium influx via TRPM2 after external stimuli triggers microglial activation and generates pro-inflammatory mediators from activated microglial cells, which further exacerbate damage. Furthermore, the TRPM2 activation process was closely related with NADPH oxidase-mediated ROS production, and PARP-1 (poly ADP-ribose polymerase-1) and ADPR (ADP-ribose) activation [60]. As mentioned in the introduction, oxidative damage following ischemic stroke is an essential factor for understanding brain damage, and is closely related with regulation of cation channel activity.

Based on our study and other previous studies, we speculated that a possible neuronal death cascade could be triggered by zinc influx via TRPM2 (Figure 7). In Figure 7A, schematic drawings describe the process of GCI-induced zinc accumulation, ROS production, and TRPM2 activation. Predicted steps of zinc-related neuronal death are as follows. (1) GCI induces vesicular zinc release from the pre-synaptic terminal to the synaptic cleft [10,13], where it translocated into post-synaptic neurons via the non-selective cation channel, TRPM2 [50,61]. In the post-synaptic neurons, translocated zinc contributes several deleterious cascades such as apoptosis [62,63], necrosis [63], as well as ROS-producing signaling. (2) Following ischemia zinc aggravates ROS production via neuronal NADPH oxidase activation [64]. Synaptically released zinc after GCI stimulates protein kinase C (PKC), and PKC contributes NADPH oxidase activation [65]. (3) A major activator of TRPM2 is ADP-ribose (ADPR), which is released upon oxidative stress including ROS [66,67]. (4) ROS-induced TRPM2 activation triggers increased secondary zinc influx [61]. (5) Secondary translocation of zinc promotes increased ROS production [59,68]. (6) Zinc and ROS induce neuronal death. In Figure 7B, the predicted mechanism of NAC on TRPM2 is shown, as follows. (7) GCI induces zinc translocation from the synaptic terminal into neurons through TRPM. Zinc increases ROS production via NADPH oxidase activation. (8) NAC-supplied cysteine enters into post-synaptic neurons through EAAC1. EAAC1 acts as a cysteine carrier in the brain. (9) Cysteine is converted to glutathione, which has an anti-oxidative effect. ROS production is attenuated by glutathione. (10) Reduction of ROS production decreases TRPM2 activation. (11) Reduced TRPM2 induces diminished secondary zinc influx. Zinc influx reduction contributes neuroprotection. (12) NAC administration induced reduced ROS production and diminished zinc influx, which finally attenuates neuronal death. Thus, as discussed above, steps (1)–(6) shown in the Figure 7A have been demonstrated by our and other group’s previous studies. However, steps (7)–(12) in the Figure 7B have been verified in this study; it is shown in Figure 2, Figure 3, Figure 4, Figure 5 and Figure 6 that NAC treatment reduced GCI-induced zinc accumulation, oxidative stress, TRPM2 activation and neuronal death.

Taken together, the present study demonstrates the efficacy of NAC in preventing GCI-induced neuronal death, sensorimotor deficits, and neurological pathology via several signaling cascades. NAC inhibits GCI-induced zinc accumulation and influx via TRPM2 into postsynaptic neurons. Therefore, we speculate that inhibition of zinc influx and TRPM2 activation by NAC may have a potential for the treatment of ischemic stroke.

## 4. Materials and Methods

### 4.1. Ethics Statement and Experimental Animals

All animal care protocols and experimental procedures were approved by the Committee on Animal Use for Research at Hallym University (protocol # Hallym 2019-70, 21 February 2020). To test our hypothesis, experimental animal groups were divided as follows: sham (Vehicle, NAC) and GCI (Vehicle, NAC). All rats were housed in a consistently controlled environment (temperature: 22 ± 2 °C, humidity: 55 ± 5%, light regulation: 12-h light/dark cycle) and given water and standard feed by Purina (Gyeonggi-do, Korea) ad libitum. To minimize any suffering of experimental animals, we used 2~3% isoflurane anesthesia during all procedures. Additionally, this manuscript was written in accordance with the standards put forth in ARRIVE (Animals in Research: Reporting In Vivo Experiments) [69].

### 4.2. Global Cerebral Ischemia Surgery

The experimental disease model of global cerebral ischemia was performed as previously described and reported [70]. To explain again in detail, rats were deeply anesthetized with 2~3% isoflurane, which was ventilated using mixed 70% nitrous oxide and 30% oxygen and kept at a body temperature of 37 ± 1 °C using a homeothermic monitoring system (Harvard Apparatus, Holliston, MA, USA). To consistently monitor systemic arterial blood pressure and to remove blood as needed, a catheter filled with 10 units of heparin was inserted into the femoral artery. Then, common carotid arteries located beside the tracheal muscle were carefully isolated and transiently occluded. In this dissecting process, we used a surgical microscope (SZ61, Olympus, Shinjuku, Japan) to enhance surgical accuracy and avoid vagus nerve impairment. At the same time, to monitor the electroencephalograph (EEG), electrodes were placed in two bilateral burr holes. To induce the ischemic condition, we drained blood (8~10 cc) from the femoral artery and set systemic arterial blood pressure within the range of 40 ± 10 mmHg. The isolated bilateral common carotid arteries were occluded with a surgical clamp (Fine Science Tools, Foster city, CA, USA). When the systemic arterial blood pressure range was within 40 ± 10 mmHg and the electroencephalograph reached a sustained isoelectric point, we maintained these conditions for 7 min to induce the global cerebral ischemic condition. After occlusion, blood circulation to the brain was restored by unclamping the device and reperfusing the removed blood. During this period, the systemic arterial blood pressure and EEG signal were consistently monitored and vital signs of the experimental animal were periodically monitored and adjusted as required. Experimental animal groups were immediately administered NAC to the intraperitoneal space following restoration of blood perfusion and normal EEG activity; vehicle groups were given the same volume of 0.9% normal saline.

### 4.3. Experimental Procedures and NAC Administration

Planned experimental procedures were 3 days and 7 days. These time points mean the termination of GCI surgery. In the acute phase (3 days), GCI-induced neurodegenerative cascades were assessed using several immunostaining processes for histological evaluation. In the chronic phase (7 days), GCI-induced cognitive decline and neurological deficits were verified by behavior tests. Vehicle and NAC was administered daily during the acute and chronic phases. To investigate the effects of NAC post-administration to reduce GCI-induced hippocampal neurodegenerations, we used NAC (Sigma-Aldrich, St. Louis, MO, USA). NAC was dissolved with 0.9% normal saline and injected into the intraperitoneal space once per day for 3 days and 7 days at a dose of 150 mg/kg in the present study. Vehicle groups (sham and GCI) were administered the same volumes of 0.9% normal saline only.

### 4.4. Sample Preparation

Three and seven days after global cerebral ischemia, all experimental rats were deeply anesthetized with urethane (1.5 g/kg) dissolved in 0.9% normal saline to obtain brain samples. For removal of whole blood, deeply anesthetized experimental animals were intracardially perfused with 0.9% normal saline followed by 4% paraformaldehyde (PFA) dissolved in phosphate-buffered saline (PBS) for brain sample fixation. Terminated sample fixation occurred during perfusion, and the whole brain sample was obtained and was immersed in 4% PFA for one hour. After post-fixation, samples were moved into 30% sucrose for cryoprotection. Brain samples sank to the bottom of the tube, and the whole brain was coronally sectioned into thicknesses of 30 μm each using a cryostat microtome (CM1850, Leica, Wetzlar, Germany).

### 4.5. Assessment of Neuronal Death

Three or seven days after GCI, we estimated hippocampal neuronal death using Fluoro-Jade B (FJB, Histo-Chem, Jefferson, AR) staining method (as previously described in detail [12]). Approximately nine to ten coronal brain sections (based on 30-μm thickness, 270- to 300-μm intervals) were obtained from 3.00 to 4.68 mm caudal to bregma. These brain sections were mounted on gelatin-coated slides, and photographed using a fluorescence microscope (SZ61, Olympus, Shinjuku, Japan, FITC green fluorescence excitation-emission wavelength: 460–490 nm). To avoid experimenter bias, a blind observer counted the whole number of FJB-positive neurons in the hippocampal subiculum, cornu ammonis 1 (CA1), and CA2 regions. Collected data were used for statistical analysis.

### 4.6. Zinc Fluorescence Staining

To verify the alteration of intraneuronal free zinc concentrations under normal or ischemic conditions, fresh brain samples were obtained without transcardial perfusion process, immediately frozen, and stored in a freezer at a consistently maintained temperature of −80 °C. These samples were coronally sectioned with 10-µm thickness in cryostat and sectioned samples were stained with *N*-(6-methoxy-8-quinolyl)-para-toluenesulfonamide (TSQ; Molecular Probes, Eugene, OR, USA) [25]. In the present study, the concentration of TSQ solution contained 4.5 µM TSQ, 140 mM sodium barbital, and 140 mM sodium acetate. The TSQ-zinc binding fluorescence signal was verified using an Olympus fluorescence microscope (DAPI blue fluorescence excitation-emission wavelength: ~400 nm).

### 4.7. Histological Analysis

Before initiation of brain tissue immunostaining, blocking of endogenous peroxidase activity is important. The previously described activity was blocked by 1.2% hydrogen peroxide incubation for 20 min at room temperature. After incubation and washing in 0.01% phosphate buffered saline, sections were incubated with several antibodies to analyze our data. In this study, the antibodies used were as follows: 4HNE (diluted 1:500, Alpha Diagnostic Intl. Inc., San Antonio, TX, USA), GFAP (diluted 1:1000, Abcam, Cambridge, UK), Iba-1 (diluted 1:500, Abcam), TRPM2 (diluted 1:400, Abcam), and NeuN (diluted 1:500, Billerica, Millipore Co., Burlington, MA, USA).

### 4.8. Behavior Outcome Assessment

Adhesive removal test: to test whether NAC restores against GCI-induced sensorimotor deficit, all of the experimental groups were evaluated using the adhesive removal test method for 7 consecutive days after GCI. This behavior test was conducted with reference to previous descriptions [13,71]. To explain again in detail, the overall evaluation process was divided into the following steps. Before initiation, rats were acclimated to the transparent testing box (size: 45 × 35 × 20 cm) for 1 min. After acclimating, two pieces of adhesive tape (1 × 1 cm) were attached to the palm of each forepaw and the experimental animal’s behavioral outcome for removal time in the testing box was observed (maximum time: 120 s). Removal time is defined as recognizing and detaching the adhesive tape on the bilateral forepaw. If the adhesive tape was detached by shaking its forepaw or bringing its forepaw to its mouth, time was recorded. This process was conducted for five trials with a minute between each trial. In the case that adhesive tapes could not be detached during the test period, removal time was considered the maximum.Modified neurological severity score: to test whether NAC administration attenuated GCI-induced neurological deficits, we used the modified neurological severity score (mNSS) method [72]. The mNSS includes several evaluations of motor (muscle status, abnormal movement), sensory (visual, tactile, and proprioceptive), balance, and reflex functionality. The mNSS assessment criteria were graded from 0 (normal, no deficits) to 18 (maximum points). To explain in detail, this functional test is based on several criteria that include raising subject by the tail and observing flexion (3 points), walking on the floor (3 points), sensory test (2 points), maintaining balance on a beam (6 points), and absence of reflex/abnormal movements (4 points) [73]. This procedure was conducted consecutively for 7 days after GCI. A higher score means a more severe condition.

### 4.9. Statistical Analysis

All data analyses were conducted using a blind test to reduce researcher bias. All quantified results in the present study were displayed as the mean value ± standard error of mean (SEM), and statistically significant differences were considered at *p* < 0.05. Behavioral data were analyzed by analysis of variance (ANOVA) using Statistical Package for the Social Sciences (SPSS, Chicago, IL, USA). Comparisons between vehicle and NAC-treated groups were conducted with the Mann–Whitney U test, and other comparisons between 4 groups and the remaining data were analyzed by the Kruskal–Wallis test with post-hoc Bonferroni correction. Non-parametric tests were chosen because of the small sample size and lack of normally distributed data.

## Figures and Tables

**Figure 1 ijms-21-06026-f001:**
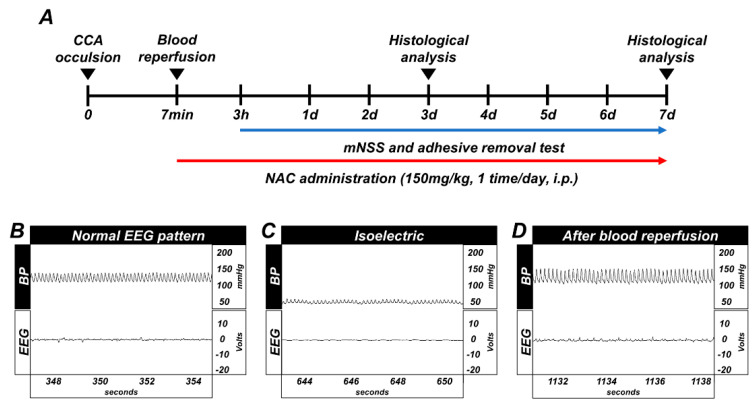
(**A**) Timeline representing the experimental design for 3 and 7 days following global cerebral ischemia (GCI). Experimental animals were given vehicle and* N*-acetyl-l-cysteine (NAC) (dosage: 150 mg/kg) once per day for 3 and 7 days. During the 7 days of the experiment, animals were subjected to behavioral outcome assessment by adhesive removal test and modified neurological severity score measurement. Global cerebral ischemia (GCI) induces electroencephalograph (EEG) and blood pressure changes, and timeline of experiment. (**B**) Normal blood pressure, resting EEG pattern. (**C**) Blood pressure decreased up to 40 (diastolic)–50 (systolic) mmHg, and EEG displayed isoelectric pattern. (**D**) Blood pressure and EEG pattern completely normalized after blood reperfusion.

**Figure 2 ijms-21-06026-f002:**
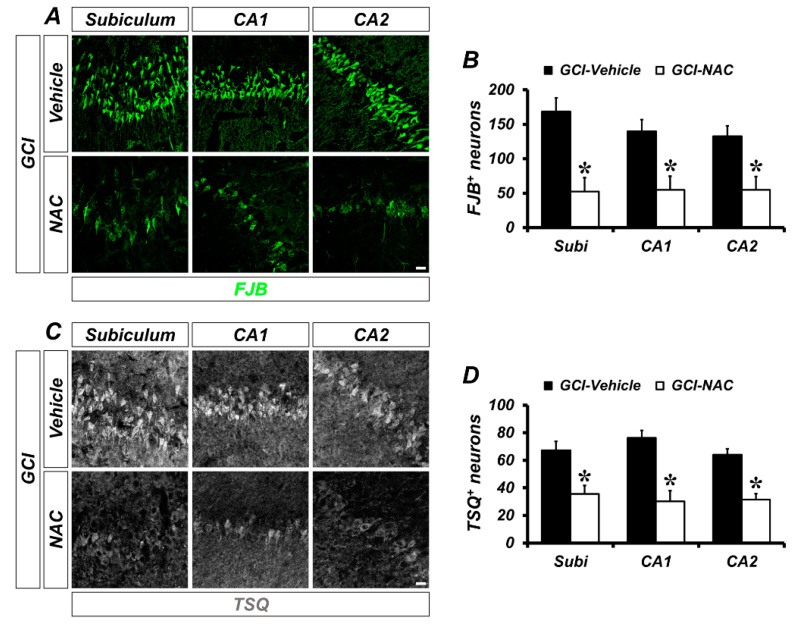
Number of GCI-induced degenerating neurons and zinc accumulation was attenuated by NAC treatment. (**A**) Brain sections were stained with fluorescence dye for detection of degenerating neurons, Fluoro-Jade B (FJB, green signal) in the hippocampal subiculum, cornus ammonis (CA) 1 and CA2 regions. Brain tissues that suffered GCI condition have numerous degenerating neurons in hippocampal regions and administration of NAC reduces the number of FJB-positive neurons in the same regions. Scale bar = 20 μm. (**B**) Quantification of FJB-positive neurons counted in each hippocampal region. Data are mean ± standard error of mean (SEM), *n* = 6 each group, * *p* < 0.05 versus vehicle-treated group (Mann–Whitney U test, Subiculum: z = 2.562, *p* = 0.010; CA1: z = 2.402, *p* = 0.016; CA2: z = 2.082, *p* = 0.037). (**C**) Fluorescence microscopic images of free zinc stained by *N*-(6-methoxy-8-quinolyl)-para-toluenesulfonamide (TSQ) staining. Bright blue fluorescence signal means ischemic condition-induced excessive zinc release and translocation into hippocampal neurons. The NAC-administered group had a remarkably reduced number of TSQ-positive neurons. Scale bar = 20 μm. (**D**) Quantification of TSQ fluorescence signal-positive neurons counted from the hippocampus. Data are mean ± SEM, *n* = 4–5 each group, * *p* < 0.05 versus vehicle-treated group (Mann–Whitney U test, Subiculum: z = 2.205, *p* = 0.027; CA1: z = 2.449, *p* = 0.014; CA2: z = 2.449, *p* = 0.014).

**Figure 3 ijms-21-06026-f003:**
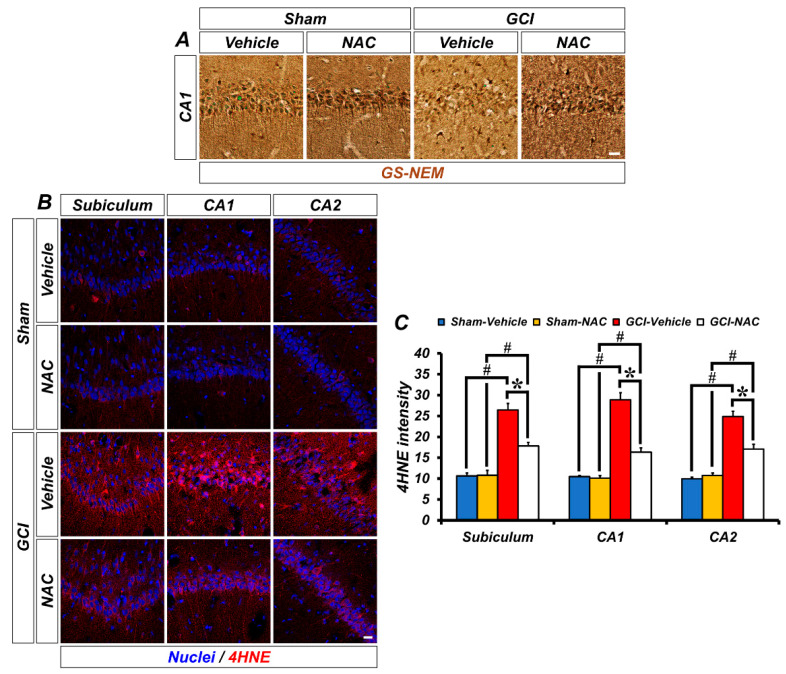
GCI-induced decreasing concentration of glutathione and increasing oxidative damage was restored by anti-oxidant, NAC. (**A**) Representative histological images show concentration of glutathione marker GS-NEM in the hippocampal CA1. Neuronal glutathione level was remarkably reduced after GCI and was restored to normal levels. Scale bar = 20 μm. (**B**) 4-hydroxy-2-nonenal (4-HNE) immunofluorescence staining was used as a marker for lipid peroxidation. Sham–vehicle and NAC groups show minimized 4-HNE-positive signal (red color) in hippocampal regions. It was remarkably increased in the GCI–vehicle group and NAC administration after GCI attenuates oxidative damage-induced lipid peroxidation. Scale bar = 20 μm. (**C**) Bar graph means intensity of 4-HNE signal in the hippocampal regions. Data are mean ± SEM, *n* = 4–7 each group, * *p* < 0.05 versus vehicle-treated group; # *p* < 0.05 versus sham-operated group (Kruskal–Wallis test followed by Bonferroni post-hoc test, Subiculum: chi square = 18.688, df = 3, *p* < 0.001; CA1: chi square = 18.783, df = 3, *p* < 0.001; CA2: chi square = 18.229, df = 3, *p* < 0.001).

**Figure 4 ijms-21-06026-f004:**
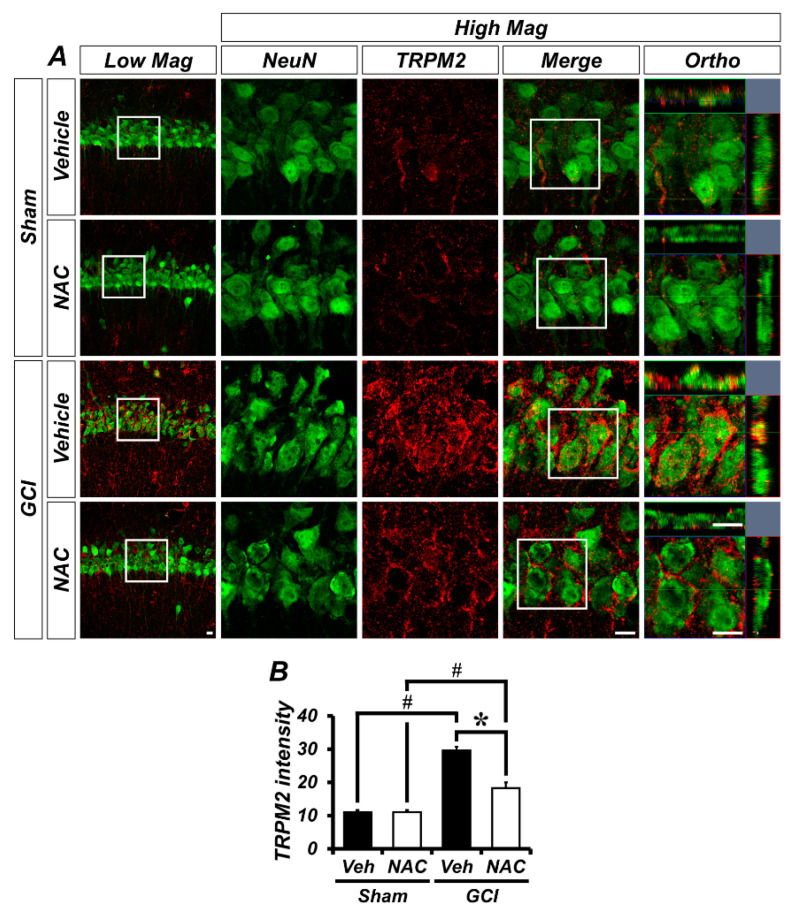
GCI-induced transient receptor potential melastatin 2 (TRPM2) activation was attenuated by NAC treatment. (**A**) Representative histological images of TRPM2-positive signal in the hippocampal CA1. The neuronal TRPM2 level was remarkably increased in the GCI–vehicle group and NAC administration after GCI reduced TRPM2 activation. Scale bar = 20 μm. (**B**) Bar graph shows intensity of TRPM2-positive signal (red color) within the hippocampal pyramidal layer. Data are mean ± SEM, *n* = 5 each group, * *p* < 0.05 versus vehicle-treated group; # *p* < 0.05 versus sham-operated group (Kruskal–Wallis test followed by Bonferroni post-hoc test: chi square = 16.097, df = 3, *p* = 0.001).

**Figure 5 ijms-21-06026-f005:**
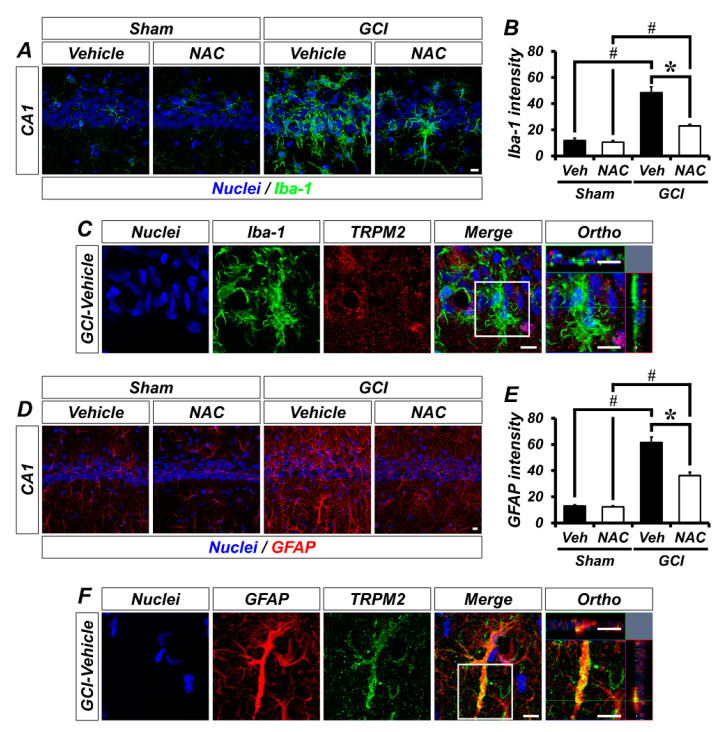
GCI-induced glial cell activation was attenuated by NAC treatment. (**A**) Microglial activation was estimated by Iba-1 immuno-staining. Sham groups (vehicle and NAC) had no difference in Iba-1-positive signal (green color). GCI triggers excessive microglial activation in the hippocampal pyramidal layer, and this was attenuated by NAC administration for 3 days after onset of GCI. Scale bar = 10 μm. (**B**) Bar graph representing intensity of Iba-1-positive signals from hippocampal pyramidal layer administered with vehicle and NAC for 3 days after sham and GCI surgery. Data are mean ± SEM, *n* = 5–7 each group, * *p* < 0.05 versus vehicle-treated group; # *p* < 0.05 versus sham-operated group (Kruskal–Wallis test followed by Bonferroni post-hoc test: chi square = 20.248, df = 3, *p* < 0.001). (**C**) Distribution of TRPM2 in microglial cells (merged image). (**D**) Astrocyte activation was evaluated by GFAP immuno-staining. Sham groups had no difference in GFAP-positive signal (red color). Astrocyte activation was stimulated by GCI, and reduced by NAC administration. Scale bar = 10 μm. (**E**) Bar graph displaying intensity of GFAP-positive signals from the hippocampal pyramidal layer in vehicle and NAC-treated groups after sham and GCI surgery. Data are mean ± SEM, *n* = 5–6 each group, * *p* < 0.05 versus vehicle-treated group; # *p* < 0.05 versus sham-operated group (Kruskal–Wallis test followed by Bonferroni post-hoc test: chi square = 18.747, df = 3, *p* < 0.001). (**F**) Distribution of TRPM2 in astrocyte following GCI (merged image).

**Figure 6 ijms-21-06026-f006:**
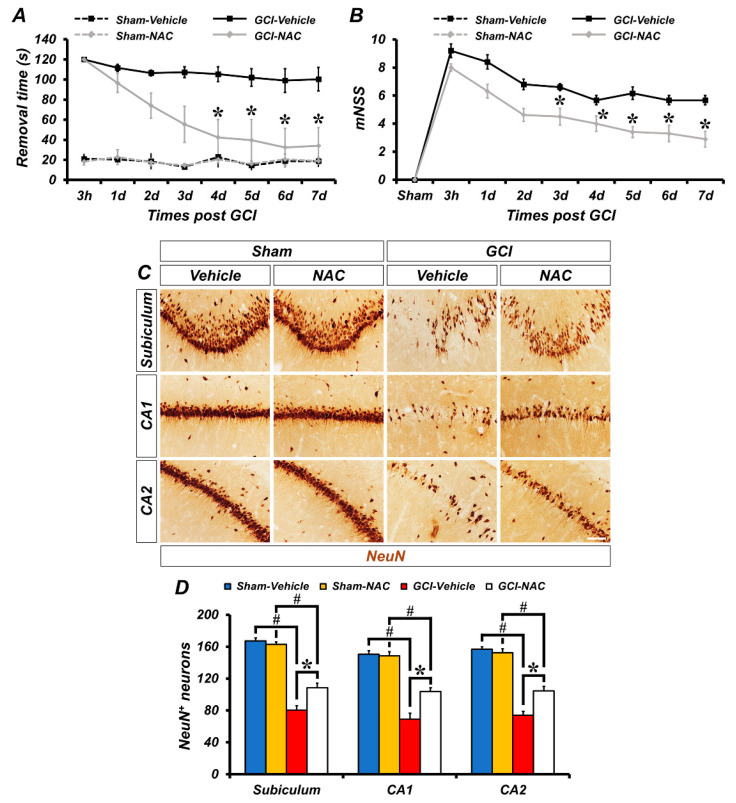
Post administration of NAC reduces GCI-induced behavioral outcome deficits and delays neuronal loss. (**A**) The adhesive removal test to assess sensorimotor deficit was conducted at both 3 h and 7 days after sham and GCI surgery. Sham groups showed no difference in time to remove the adhesive tape during the test. GCI–vehicle group took significantly longer to remove the tape. Post treatment of NAC after GCI reduces removal time (time dependently). Data are mean ± SEM, *n* = 5–8 each group, * *p* < 0.05 versus vehicle-treated group; # *p* < 0.05 versus sham-operated group (repeated measure test followed by ANOVA, time: F = 6.365, *p* < 0.001; group: F = 33.354, *p* < 0.001; time * group: F = 3.061, *p* < 0.001). (**B**) The mNSS was conducted at 3 h and 7 consecutive days after sham and GCI surgery. Sham groups indicate a 0 score which means all tasks were performed completely without defect. Under the ischemic condition, the vehicle group showed a failure of mNSS sessions. Post administration, NAC improved mNSS procedures and reduced failure of tasks. Data are mean ± SEM, *n* = 6–8 each group, * *p* < 0.05 versus the vehicle-treated group (repeated measure test followed by ANOVA, time: F = 35.621, *p* < 0.001; group: F = 17.576, *p* = 0.001; time * group: F = 2.322, *p* = 0.032). (**C**) Immunohistochemistry images show NeuN-positive neurons in the hippocampal subiculum, CA1, and CA2 regions. Scale bar = 50 μm. (**D**) Bar graph shows analysis of NeuN-positive cells in each hippocampal region. Post treatment of NAC restores GCI-induced delayed neuronal loss in the following 7 days. Data are mean ± SEM, *n* = 5–6 each group, * *p* < 0.05 versus vehicle-treated group; # *p* < 0.05 versus sham-operated group (Kruskal–Wallis test followed by Bonferroni post-hoc test, Subiculum: chi square = 17.792, df = 3, *p* < 0.001; CA1: chi square = 17.958, df = 3, *p* < 0.001; CA2: chi square = 17.735, df = 3, *p* < 0.001).

**Figure 7 ijms-21-06026-f007:**
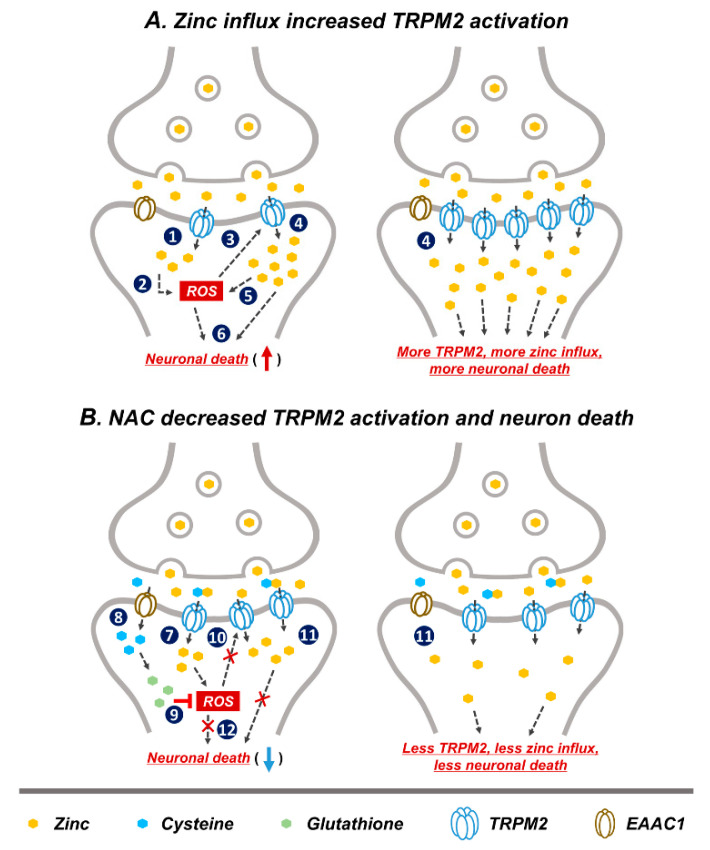
Predicted reactions of NAC post treatment on GCI-induced neuronal death. Representative schematic illustration indicates NAC action via reduction of reactive oxygen species (ROS) production and subsequent decrease in TRPM2 activation. (**A**) (1) Released zinc after GCI moves into the post-synaptic neurons through TRPM2. (2) Translocated zinc contributes to ROS production. (3) Oxidative damage including ROS activated TRPM2. (4) ROS-induced TRPM2 activation increased secondary zinc influx. (5) Increased translocated of zinc contributes to greater ROS production. (6) Zinc and ROS induces neuronal death. (1)–(6) Image shows the hypothesized cascade produced upon GCI-induced zinc influx and TRPM2 activation. (**B**) (7) Zinc translocated into the post-synaptic neurons via TRPM2 after GCI. (8) Cysteine supplied by NAC administration moves into post-synaptic neurons through excitatory amino acid carrier 1 (EAAC1). (9) Cysteine is converted to glutathione which has an anti-oxidant effect and inhibits ROS production. (10) Inhibition of ROS production leads to reduced TRPM2 activation. (11) Reduced TRPM2 activation leads to less secondary zinc influx. (12) Reduction of ROS production and secondary zinc influx reduces neuronal death. (7)–(12) Process shows possible neuroprotective cascade of NAC after GCI.

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
