# Peer review of "Transient Receptor Potential Melastatin 2 (TRPM2) Inhibition by Antioxidant, N-Acetyl-l-Cysteine, Reduces Global Cerebral Ischemia-Induced Neuronal Death"

_ijms, 2020, doi:10.3390/ijms21176026_

Round 1

Reviewer 1 Report

The study entitled “Transient Receptor Potential Melastatin 2 (TRPM2) Inhibition by Antioxidant, N-Acetyl-L-cysteine, Reduces Global Cerebral Ischemia-induced Neuronal Death” investigated the effect of NAC for treating global cerebral ischemia. They show that NAC protected against global cerebral ischemia, associated with the reduction of zinc, lipid peroxidation, TRPM2. However, the novelty of this study is not high, as the therapeutic effects of antioxidants on global ischemia are well-known. The detrimental effect of TRPM2 in a stroke model is also well established. It is also known that TRPM2 can be triggered by several factors, including reactive oxygen species (ROS).

Furthermore, the current study design only shows the associated phenomenons, and the mechanistic study is descriptive. In other words, the TRPM2 reduction may be a result, rather than the cause of reduced brain injury by NAC. Therefore, the conclusion proposed by authors seems not valid.

Author Response

Reviewer #1: The study entitled “Transient Receptor Potential Melastatin 2 (TRPM2) Inhibition by Antioxidant, N-Acetyl-L-cysteine, Reduces Global Cerebral Ischemia-induced Neuronal Death” investigated the effect of NAC for treating global cerebral ischemia. They show that NAC protected against global cerebral ischemia, associated with the reduction of zinc, lipid peroxidation, TRPM2. However, the novelty of this study is not high, as the therapeutic effects of antioxidants on global ischemia are well-known. The detrimental effect of TRPM2 in a stroke model is also well established. It is also known that TRPM2 can be triggered by several factors, including reactive oxygen species (ROS). Furthermore, the current study design only shows the associated phenomenons, and the mechanistic study is descriptive. In other words, the TRPM2 reduction may be a result, rather than the cause of reduced brain injury by NAC. Therefore, the conclusion proposed by authors seems not valid.

<Response: We appreciate your comments and agree with this reviewer’s concerns. In the present study, we wanted to demonstrate that global cerebral ischemia-induced neuronal death can be induced by the following chain reaction of zinc translocation, ROS production and TRPM2 activation, which can be attenuated by NAC supplementation. We described it in detail in Figure 7 and discussion section.

  1. Released vesicular zinc after global cerebral ischemia was translocated into the post-synaptic neurons via TRPM2.
  2. Zinc increased ROS production via NADPH oxidase activation [1].
  3. Oxidative stress including ROS activated TRPM2 [2,3].
  4. ROS-induced TRPM2 activation secondary increased zinc influx into neurons.
  5. Increased zinc influx aggravated more ROS
  6. Zinc and ROS induced neuronal death.

1~6 process is the possible cascade of zinc influx-related TRPM2 activation after global cerebral ischemia.

Next, we speculate the role of NAC on TRPM2.

  1. Zinc is translocated into neurons after global cerebral ischemia.
  2. NAC supplied cysteine, entered into post-synaptic neurons via excitatory amino acid carrier 1 (EAAC1).
  3. Cysteine is converted to glutathione, which decreases ROS
  4. Inhibition of ROS leading to less TRPM2 activation.
  5. Less TRPM2 induced less secondary zinc influx.
  6. Less ROS and less free zinc attenuated neuronal death.

7~12 process is the possible neuroprotective cascade of NAC after global cerebral ischemia.

References

  1. Suh, S.W.; Gum, E.T.; Hamby, A.M.; Chan, P.H.; Swanson, R.A. Hypoglycemic neuronal death is triggered by glucose reperfusion and activation of neuronal NADPH oxidase. J Clin Invest 2007, 117, 910-918, doi:10.1172/JCI30077.
  2. Li, X.; Jiang, L.H. A critical role of the transient receptor potential melastatin 2 channel in a positive feedback mechanism for reactive oxygen species-induced delayed cell death. J Cell Physiol 2019, 234, 3647-3660, doi:10.1002/jcp.27134.
  3. Di, A.; Gao, X.P.; Qian, F.; Kawamura, T.; Han, J.; Hecquet, C.; Ye, R.D.; Vogel, S.M.; Malik, A.B. The redox-sensitive cation channel TRPM2 modulates phagocyte ROS production and inflammation. Nat Immunol 2011, 13, 29-34, doi:10.1038/ni.2171.

Reviewer 2 Report

In this manuscript, the authors did in vivo study to investigate whether anti-oxidant treatment, such as with N-acetyl-l-cysteine (NAC), provides neuroprotection via regulation of TRPM2, following global cerebral ischemia (GCI). They suggested that treating rats with NAC (150mg/kg/day) for 3 and 7 days, before sacrifice could lead to reduced activation of GCI-induced neuronal death cascades, such as lipid peroxidation, microglia and astroglia activation, free zinc accumulation, as well as TRPM2 over-activation. This manuscript is well written with a good standard of English language. Manuscript is very interesting and useful for readers and researchers which operate in the biomedical field. A good scientific and experimental exposure of the study. The methodologies are appropriate and aligned with the proposed objectives.

Author Response

<Response: We appreciate this reviewer’s comments.>

Reviewer 3 Report

The manuscript titled as ‘Transient Receptor Potential Melastatin 2 (TRPM2) Inhibition by
Antioxidant, N-Acetyl-L-cysteine, Reduces Global Cerebral Ischemia-induced
Neuronal Death’ by Hong et al. verified whether NAC administration attenuates GCI-induced neuronal death via inhibition of TRPM2 channels though GCI brain disease model in adult rats. The authors conclude that NAC may be an outstanding therapeutic agent for preventing GCI-induced neuronal death, and inhibition of TRPM2 channel by NAC may have therapeutic potential for the treatment of ischemic stroke. This work should be of wide interests to most researchers on neuroscience and molecular medicine etc.

This manuscript has sufficient novel and findings and the method described is highly practical. I recommend that the manuscript be accepted with some revisions. The following points need to be addressed:

  1. The authors did not explain why they use a dose of 150 mg/kg for NAC administration, any more backgrounds?
  2. What is the action mechanism of N-Acetyl-L-cysteine on TRPM2?

Author Response

Reviewer #3: The manuscript titled as ‘Transient Receptor Potential Melastatin 2 (TRPM2) Inhibition by Antioxidant, N-Acetyl-L-cysteine, Reduces Global Cerebral Ischemia-induced Neuronal Death’ by Hong et al. verified whether NAC administration attenuates GCI-induced neuronal death via inhibition of TRPM2 channels though GCI brain disease model in adult rats. The authors conclude that NAC may be an outstanding therapeutic agent for preventing GCI-induced neuronal death, and inhibition of TRPM2 channel by NAC may have therapeutic potential for the treatment of ischemic stroke. This work should be of wide interests to most researchers on neuroscience and molecular medicine etc. This manuscript has sufficient novel and findings and the method described is highly practical. I recommend that the manuscript be accepted with some revisions. The following points need to be addressed:

1. The authors did not explain why they use a dose of 150 mg/kg for NAC administration, any more backgrounds?

<Response: We appreciate this reviewer’s comments. Several previous studies showed that NAC treatment (150 mg/kg) resulted in sustained neuroprotection and neurological recovery after transient global cerebral ischemia or traumatic brain injury [1-4]>

2. What is the action mechanism of N-Acetyl-L-cysteine on TRPM2?

<Response: We speculated that the action mechanism of N-Acetyl-L-cysteine on TRPM2 after global cerebral ischemia (GCI) as follows.

(1) Released zinc from pre-synaptic vesicles after GCI entered into post-synaptic neurons through TRPM2, and then increased ROS production via NADPH oxidase activation.

(2) N-Acetyl-L-cysteine residue cysteine, entered into post-synaptic neurons through excitatory amino acid carrier 1 (EAAC1).

(3) Cysteine produced anti-oxidant, glutathione and decreased ROS production.

(4) ROS-triggered TRPM2 activation was attenuated by glutathione.

(5) Less TRPM2 activation leading to less zinc influx.

(6) Decreased ROS production and TRPM2 activation reduces neuronal death.

References

  1. Jang, B.G.; Won, S.J.; Kim, J.H.; Choi, B.Y.; Lee, M.W.; Sohn, M.; Song, H.K.; Suh, S.W. EAAC1 gene deletion alters zinc homeostasis and enhances cortical neuronal injury after transient cerebral ischemia in mice. J Trace Elem Med Biol 2012, 26, 85-88, doi:10.1016/j.jtemb.2012.04.010.
  2. Choi, B.Y.; Kim, J.H.; Kim, H.J.; Lee, B.E.; Kim, I.Y.; Sohn, M.; Suh, S.W. EAAC1 gene deletion increases neuronal death and blood brain barrier disruption after transient cerebral ischemia in female mice. Int J Mol Sci 2014, 15, 19444-19457, doi:10.3390/ijms151119444.
  3. Holmay, M.J.; Terpstra, M.; Coles, L.D.; Mishra, U.; Ahlskog, M.; Oz, G.; Cloyd, J.C.; Tuite, P.J. N-Acetylcysteine boosts brain and blood glutathione in Gaucher and Parkinson diseases. Clin Neuropharmacol 2013, 36, 103-106, doi:10.1097/WNF.0b013e31829ae713.
  4. Khan, M.; Sekhon, B.; Jatana, M.; Giri, S.; Gilg, A.G.; Sekhon, C.; Singh, I.; Singh, A.K. Administration of N-acetylcysteine after focal cerebral ischemia protects brain and reduces inflammation in a rat model of experimental stroke. J Neurosci Res 2004, 76, 519-527, doi:10.1002/jnr.20087.

Reviewer 4 Report

The paper as a whole makes a good impression and is of great interest to the reader.
 Small disadvantages include the following:

  1. In fig. 3C Sham-NAC and GCl-Vehicle groups are marked with almost the same color. I recommend using different colors in the histogram to represent these groups.
    2. What were the authors guided by in choosing the criteria for statistical analysis? Why, in a comparative assessment of the number of FGB and TSQ cells in the hippocampus, the authors used Mann-Whitney U test (Fig.2B and D), and when assessing the 4HNE-intensity Kruskal-Wallis test followed by Bonferroni post-hoc test (Fig.3C).
    3. It is necessary to increase the number of references in the last 5 years. Their share should be about half of all references.
    4. Correction of English with natural speakers is required.

Author Response

Reviewer #4: The paper as a whole makes a good impression and is of great interest to the reader. Small disadvantages include the following:

1. In fig. 3C Sham-NAC and GCl-Vehicle groups are marked with almost the same color. I recommend using different colors in the histogram to represent these groups.

<Response: We corrected it in the revised manuscript.>

2. What were the authors guided by in choosing the criteria for statistical analysis? Why, in a comparative assessment of the number of FGB and TSQ cells in the hippocampus, the authors used Mann-Whitney U test (Fig.2B and D), and when assessing the 4HNE-intensity Kruskal-Wallis test followed by Bonferroni post-hoc test (Fig.3C).

<Response: In the present study, we used the Mann-Whitney U test to compare differences between two independent groups and analyzed the data by the non-parametric Kruskal Wallis test with post hoc analysis using Bonferroni correction to compare the values among 4 groups. The Mann-Whitney U test is more commonly used for comparing two groups (the non-parametric version of the student t-test). The Kruskal-Wallis test is a non-parametric approach to the one-way ANOVA for comparing more than two groups. The reason to use non-parametric tests was that the number of cases was small and the values were not normally distributed. >

3. It is necessary to increase the number of references in the last 5 years. Their share should be about half of all references.

<Response: We updated the references in the revised manuscript.>

4. Correction of English with natural speakers is required.

<Response: We corrected it. This revised manuscript is edited by MDPI English Editing company. We attached editing certificate.>

Round 2

Reviewer 1 Report

This study shows that NAC treatment provided neuroprotective effects in a global cerebral ischemia model, which was associated with the reduction of oxidative stress, Zinc and TRPM2 level. However, this is a descriptive study that does not address the hypothesis: TRPM2 regulates Zinc for neuroprotection. It needs much more evidence to support this hypothesis. The current study just demonstrated the beneficial effects of NAC, which is largely predictable.

Author Response

Response: We appreciate this reviewer’s comment. As this reviewer’s point, the present study has some limitations which have to be pointed out. We understand this reviewer’s concerns, however, this study based on in vivo experiments and it is hard to distinguish the precise mechanism. We wish this reviewer understand this matters. But, in this revision, we added additional sentences and references for each steps that based on other groups’ findings as well as our present results to clarify our hypothesis. Further experiments will be needed to understand the effectiveness of NAC for reduction of zinc influx via TRPM2.